# CD47 cross-dressing by extracellular vesicles expressing CD47 inhibits phagocytosis without transmitting cell death signals

Yang Li[1,2], Yan Wu[2], Elena A Federzoni[3], Xiaodan Wang[1], Andre Dharmawan[3], Xiaoyi Hu[2], Hui Wang[2], Robert J Hawley[2], Sean Stevens[3], Megan Sykes[2], Yong-Guang Yang[1,4]*

[1]Key Laboratory of Organ Regeneration and Transplantation of the Ministry of Education, The First Hospital, and Institute of Immunology, Jilin University, Changchun, China; [2]Columbia Center for Translational Immunology, Columbia University Medical Center, New York, United States; [3]Lung Biotechnology PBC, Silver Spring, United States; [4]International Center of Future Science, Jilin University, Changchun, China

*For correspondence: yongg@jlu.edu.cn

Competing interest: The authors declare that no competing interests exist.

**Abstract** Transgenic CD47 overexpression is an encouraging approach to ameliorating xenograft rejection and alloresponses to pluripotent stem cells, and the efficacy correlates with the level of CD47 expression. However, CD47, upon ligation, also transmits signals leading to cell dysfunction or death, raising a concern that overexpressing CD47 could be harmful. Here, we unveiled an alternative source of cell surface CD47. We showed that extracellular vesicles, including exosomes, released from normal or tumor cells overexpressing CD47 (transgenic or native) can induce efficient CD47 cross-dressing on pig or human cells. Like the autogenous CD47, CD47 cross-dressed on cell surfaces is capable of interacting with SIRPα to inhibit phagocytosis. However, ligation of the autogenous, but not cross-dressed, CD47 induced cell death. Thus, CD47 cross-dressing provides an alternative source of cell surface CD47 that may elicit its anti-phagocytic function without transmitting harmful signals to the cells. CD47 cross-dressing also suggests a previously unidentified mechanism for tumor-induced immunosuppression. Our findings should help to further optimize the CD47 transgenic approach that may improve outcomes by minimizing the harmful effects of CD47 overexpression.

## Editor's evaluation

This study by Li et al., describes an important molecular mechanism that limits the rejection of xeno-transplant or allogenic tissue. By presenting compelling evidence, that describes a mechanism to cross-dress xenogeneic cells with human CD47. Li et al., investigated if this process can improve grafted tissue survival by halting phagocytosis of xenogeneic cells by host human monocytes while also diminishing SIRP-a mediated apoptosis. This work will be of interest to basic and transplant immunology.

## Introduction

CD47 is ubiquitously expressed and acts as a ligand of signaling regulatory protein (SIRP)α, a critical inhibitory receptor on macrophages and dendritic cells (DCs). Emerging evidence indicates that the

CD47-SIRPα signaling pathway plays an important role in regulation of macrophage and DC activation, offering a promising intervention target for immunological disorders. CD47KO cells are vigorously rejected by macrophages after infusion into syngeneic wild-type (WT) mice, demonstrating that CD47 provides a 'don't eat me' signal to macrophages (*Oldenborg et al., 2000*; *Wang et al., 2007a*). Xeno-transplantation using pigs as the transplant source has the potential to resolve the severe shortage of human organ donors, a major limiting factor in clinical transplantation (*Yang and Sykes, 2007*). We reported that the strong rejection of xenogeneic cells by macrophages (*Abe et al., 2002*) is largely caused by the lack of functional interaction between donor CD47 and recipient SIRPα (*Wang et al., 2007b*, *Ide et al., 2007*; *Navarro-Alvarez and Yang, 2014*). These findings led to the development of human CD47 transgenic pigs that have achieved encouraging results in pig-to-nonhuman primate xenotransplantation (*Tena et al., 2017*; *Nomura et al., 2020*; *Watanabe et al., 2020*). In addition to macrophages, a sub-population of DCs also expresses SIRPα (*Wang et al., 2007a*, *Guilliams et al., 2016*). Importantly, CD47-SIRPα signaling also inhibits DC activation and their ability to prime T cells, and plays an important role in induction of T cell tolerance by donor-specific transfusion (DST) and hepatocyte transplantation (*Wang et al., 2007a*, *Wang et al., 2014*; *Zhang et al., 2016*). Thus, trans-genic expression of human CD47 in pigs may also attenuate xenoimmune responses by ameliorating DC activation and antigen presentation. More recently, transgenic overexpression of CD47 was also applied for reducing allogenicity and generating hypoimmunogenic pluripotent stem cells (*Han et al., 2019*; *Deuse et al., 2019*).

It has become increasingly evident that the CD47-SIRPα pathway plays a critical role in containing anti-tumor immune responses. CD47 upregulation was detected in various cancer cells, serving a powerful mechanism of evading macrophage killing (*Jaiswal et al., 2009*; *Chan et al., 2009*). Accordingly, treatment with CD47 blockade could inhibit tumor growth via macrophage-mediated mechanism (*Jaiswal et al., 2009*; *Willingham et al., 2012*; *Weiskopf et al., 2016*; *Chao et al., 2010*; *Liu et al., 2015a*). More recently, the antitumor activity of CD47 blockade was found to be associated with CD11c[+] DC activation and largely T cell-dependent (*Liu et al., 2015b*, *Chen et al., 2020*; *Li et al., 2020*). Taken together, these studies revealed clearly that the CD47-SIRPα pathway provides a powerful negative regulation for both innate and adaptive immune responses and is increasingly considered as an effective intervention target for protecting against transplant rejection and unleashing immune responses to cancer.

Although negative regulation of immune responses is predominantly mediated by inhibitory CD47-SIRPα signaling in macrophages and DCs, it remains largely unknown how transgenic CD47 on pig or human pluripotent stem cells and upregulated CD47 on tumor cells interact with its receptor and ligands. In the present study, we identified an alternative source of cell surface CD47. We found that extracellular vesicles (EVs), including exosomes (Exos) from cells transgenically overexpressing CD47 or tumor cells overexpressing endogenous CD47, could induce CD47 cross-dressing on pig or human cells. CD47 cross-dressed on cell surfaces can interact with SIRPα to inhibit phagocytosis. However, unlike the autogenous CD47 that, upon ligation, induces cell apoptosis and senescence (*Mateo et al., 1999*; *Gao et al., 2016*; *Meijles et al., 2017*), ligation of CD47 cross-dressed on cell surfaces is not harmful to cells. This study provides deeper insight into the effect of CD47 overexpression, which needs to be considered when designing strategies for gene-editing in pigs for xenotransplantation or in human pluripotent stem cells for cell replacement therapy, and for developing CD47 blockade-based cancer immunotherapy.

## Results

### Transgenic hCD47 cross-dressing in pig cells

CD47 cross-dressing was first identified by detecting hCD47 on pig cells that was cocultured with pig cells expressing transgenic hCD47. Two different hCD47 isoforms were used to ensure that findings are not isoform-specific. In these experiments, cell cocultures were performed using cell line cells derived from porcine aortic cells (PAOC; *Figure 1—figure supplement 1*). First, we cocultured parental PAOCs (expressing pig CD47; referred to as PAOC/CD47$^p$) with PAOCs that were genetically modified to express hCD47 isoform 2 (referred to as PAOC/CD47$^{p/h2}$) or isoform 4 (referred to as PAOC/CD47$^{p/h4}$). Two different CD47 isoforms were used to confirm the findings are not isoform-specific. Flow cytometry analysis using anti-CD47 antibodies recognizing both human and pig CD47

revealed that PAOC/CD47$^{p/h2}$ or PAOC/CD47$^{p/h4}$ cells expressed a markedly increased level of CD47 compared to PAOC/CD47$^p$ cells, and that PAOC/CD47$^p$ cells showed significantly increased CD47 staining after coculture with PAOC/CD47$^{p/h2}$ or PAOC/CD47$^{p/h4}$ (*Figure 1A*). These results suggest that PAOC/CD47$^p$ cells were cross-dressed by CD47, likely transgenic hCD47, from PAOC/CD47$^{p/h}$ cells during cultures. To confirm this possibility, we made CD47-defficient PAOC cells (via targeted deletion using CRISPR-Cas9- technology; referred to as PAOC/CD47$^{null}$) and PAOC47$^{null}$ cells that express transgenic hCD47 isoform 2 (PAOC/CD47$^{h2}$) or isoform 4 (PAOC/CD47$^{h4}$). When the PAOC cell line cells were cocultured for 24 hr, we found that PAOC47$^{null}$ cells became positively stained by both anti-h/pCD47 (*Figure 1B*, **top**) and anti-hCD47 (*Figure 1B*, **bottom**) antibodies. To rule out the possibility that, during the coculture, the CD47$^{null}$ cells did not become CD47$^+$, but PAOC/CD47$^{h2}$ cells reduced hCD47 expression, we labeled PAOC/CD47$^{null}$ (*Figure 1C*, **Left**) or PAOC/CD47$^{h2}$ (*Figure 1C*, **Right**) cells with florescence Celltrace violet and then cocultured the labeled cells with unlabeled PAOC/CD47$^{h2}$ or PAOC/CD47$^{null}$ cells, respectively. This experiment, in which fluorescence-labeling allowed for better distinguishing between the two cell populations in the cocultures, further confirmed that PAOC/CD47$^{null}$ cells can be cross-dressed by CD47 after coculture with PAOC/CD47$^{h2}$ cells (*Figure 1C*).

PAOC cells express SIRPα (*Figure 1—figure supplement 2*, **left**), and pig SIRPα is reported to interact with human CD47 (*Boettcher et al., 2019*). Furthermore, previous studies have shown that CD47-lentiviruses are more effective in transducing SIRPα$^+$ non-phagocytic tumor cells, suggesting that the interaction of CD47 with SIRPα may facilitate lentiviral engulfment by target cells (*Sosale et al., 2016*). Thus, to determine whether hCD47 cross-dressing is mediated by binding of hCD47 to pig SIRPα, we performed cocultures with pig lymphoma cell line (LCL) cells that do not express SIRPα (*Figure 1—figure supplement 2*, **right**). LCL cells cocultured with LCL cells that express hCD47 (LCL/CD47$^{p/h}$) (*Ide et al., 2007*; *Wang et al., 2011*), but not those cultured alone or mixed with LCL/CD47$^{p/h}$ immediately prior to flow cytometry analysis, were positively stained by anti-hCD47 antibodies (*Figure 1D*). Although we cannot rule out of the role of SIRPα in hCD47 cross-dressing, our data indicate that hCD47 cross-dressing can occur in an SIRPα-independent manner.

We then wished to determine if cells other than the PAOC cell lines could be a source of CD47 for cross-dressing. Toward this end, PAOC47$^{null}$ cells were cocultured with bone marrow cells (BMCs) from hCD47-transgenic miniature swine. Like pig cells cocultured with hCD47-transgenic cell lines (*Figure 1A–D*), PAOC47$^{null}$ cells also became positively stained by anti-hCD47 antibodies after being cocultured with hCD47-tg swine cells (*Figure 1E*).

## Cross-dressing with native CD47 from human T cell leukemia cells

We next determined whether cells can be cross-dressed by native CD47 using human T-cell leukemia Jurkat cells that express a higher level of CD47 than normal hematopoietic cells (*Figure 2A*). In order to clearly identify cross-dressed CD47 on cell surface, CD47-defficient Jurkat cells were generated using the CRISPR-Cas9 technique (*Figure 2—figure supplement 1*) and cocultured for 24 hr with the parental WT Jurkat cells or with hCD47-tg pig LCL cells (*Figure 2B*). Flow cytometry analysis revealed that CD47KO Jurkat cells were clearly stained positive by anti-hCD47 antibodies after coculture with parental WT Jurkat cells or hCD47-tg pig LCL cells compared to those cultured alone or mixed immediately before staining (*Figure 2B*). Furthermore, pig LCL cells also became positive for human CD47 staining after coculture for 24 hr with WT Jurkat cells (*Figure 2B*). More than half of the cross-dressed hCD47 (determined by MFI) remained on pig LCL cells 3 days after being separated from hCD47-expressing cells (*Figure 2C*), indicating that hCD47 cross-dressing was relatively stable. We also noted that not only hCD47, but other membrane proteins of WT Jurkat cells were also cross-dressed on pig LCL cells (*Figure 2—figure supplement 2*). While to a lower level than untreated cells, hCD47 cross-dressing was clearly detected on LCL cells that were pretreated with cytochalasin D, a cell-permeable potent inhibitor that disrupts actin microfilaments (*Cooper, 1987*), after cocultured with cytochalasin D-pretreated WT Jurkat cells (*Figure 2—figure supplement 3A*), indicating that a functional cytoskeleton is not absolutely required for, but may facilitate hCD47 cross-dressing. These results indicate that human CD47 cross-dressing could be induced by not only hCD47-transgenic cells but also tumor cells that express only the native CD47.

**Figure 1**

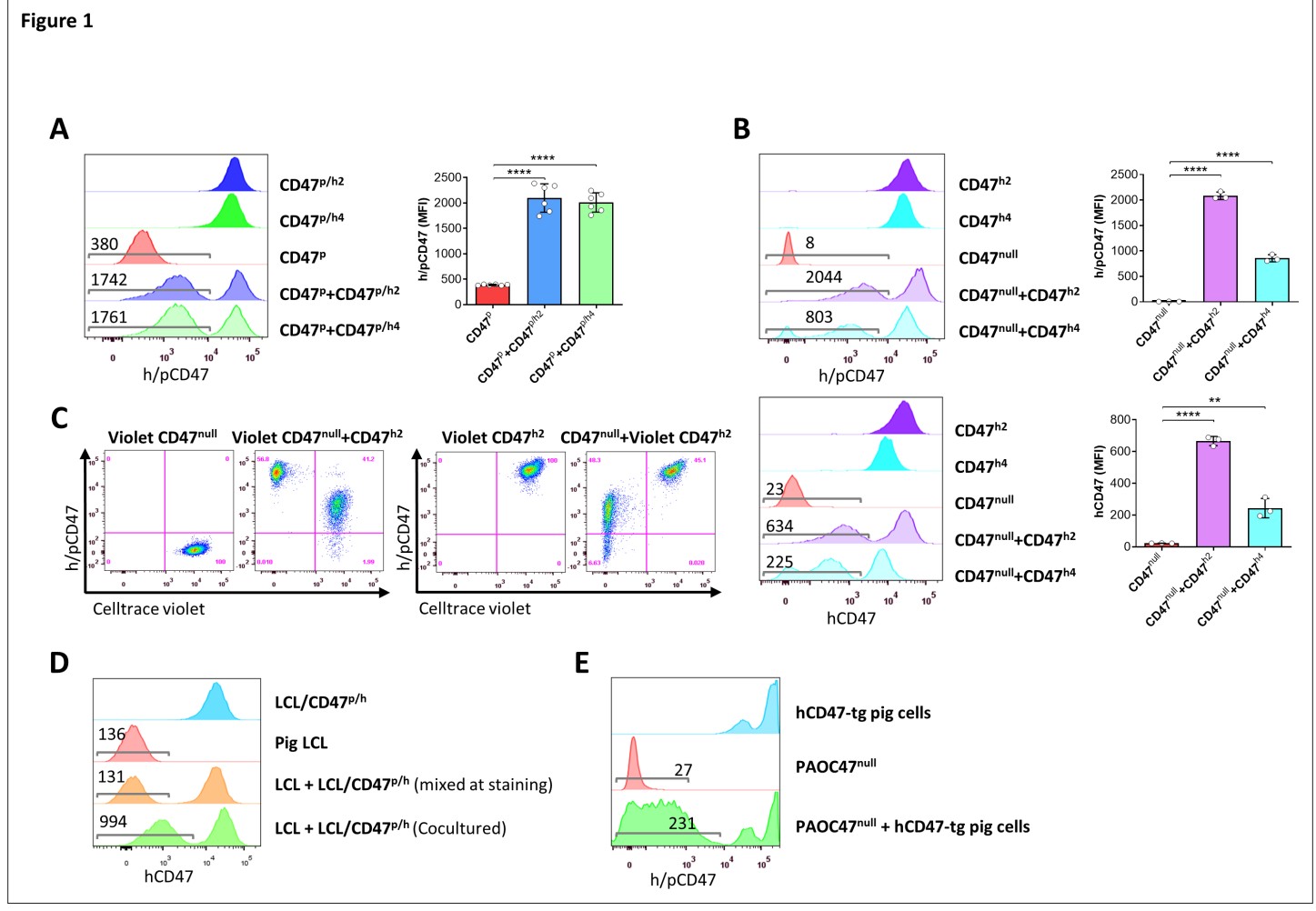

**Figure 1.** Transgenic hCD47 cross-dressing on pig cells. (**A**) Porcine aortic cell (PAOC)/CD47$^p$ cells and PAOC/CD47$^{p/h2}$ or PAOC/CD47$^{p/h4}$ cells were cultured alone or cocultured for 24 hr, and hCD47 cross-dressing on gated PAOC/CD47$^p$ cells was assessed by flow cytometry using anti-h/pCD47-PE mAb (clone CC2C6, reacting with both human and pig CD47). Shown are representative histogram profiles (left; the numbers in the figure indicate the median fluorescent intensity [MFI] of gated PAOC/CD47$^p$ cells), and average MFI (right; mean ± SDs; n=6 replicates per group) of gated PAOC/CD47$^p$ cells in the indicated cell cultures. ****, p<0.0001 (two-tailed unpaired t-test). Results shown are representative of three independent experiments. (**B**) PAOC/CD47$^{null}$, and PAOC/CD47$^{h2}$ or PAOC/CD47$^{h4}$ were cultured alone or cocultured for 24 hr, and analyzed for hCD47 cross-dressing on gated PAOC/CD47$^{null}$ cells by flow cytometry using anti-h/pCD47-PE mAb (*top*) or anti-hCD47-BV786 mAb (*bottom*). Shown are representative histogram profiles (left panel; the numbers in the figure indicate the MFI of gated PAOC/CD47$^{null}$ cells), and average MFI (right panel; mean ± SDs; n=3 replicates per group) of gated PAOC/CD47$^{null}$ cells in the indicated cell cultures. **, p<0.01; ****, p<0.0001 (two-tailed unpaired t-test). Results shown are representative of three independent experiments. (**C**) Celltrace violet-labeled PAOC/CD47$^{null}$ (left panel) or PAOC/CD47$^{h2}$ (right panel) was cultured alone (Violet CD47$^{null}$ or Violet CD47$^{h2}$) or cocultured with unlabeled PAOC/CD47$^{h2}$ (Violet CD47$^{null}$ + CD47$^{h2}$) or PAOC/CD47$^{null}$ (CD47$^{null}$ or Violet CD47$^{h2}$), respectively, then the cells were stained by anti-h/pCD47-PE mAb. Shown are representative flow cytometry profiles (n=3 replicates). Results shown are representative of two independent experiments. (**D**) Pig lymphoma cell line (LCL) and hCD47-tg LCL (LCL/CD47$^{p/h}$) cells were cultured alone or cocultured for 24 hr, and analyzed for hCD47 cross-dressing on gated LCL cells (the numbers in the figure indicate the MFI of gated LCL cells). The staining control of 'mixed at staining' indicates the two types of cells were cultured separately and mixed immediately prior anti-CD47 staining. Two independent experiments were performed, and each experiment had two replicates per group. Representative flow cytometry profiles are shown. (**E**) PAOC47$^{null}$ cells were cocultured with bone marrow cells from hCD47-tg miniature swine for 2 days and analyzed for hCD47 cross-dressing on gated PAOC47$^{null}$ cells by flow cytometry using anti-h/pCD47-PE mAb (the numbers in the figure indicate the MFI of gated PAOC/CD47$^{null}$ cells). Two independent experiments were performed, and each experiment had two replicates per group. Representative flow cytometry profiles are shown.

The online version of this article includes the following figure supplement(s) for figure 1:

**Figure supplement 1.** Generation of CD47-deficient and hCD47-trangeneic porcine aortic cell (PAOC) cell lines.

**Figure supplement 2.** SIRPα expression on pig cells.

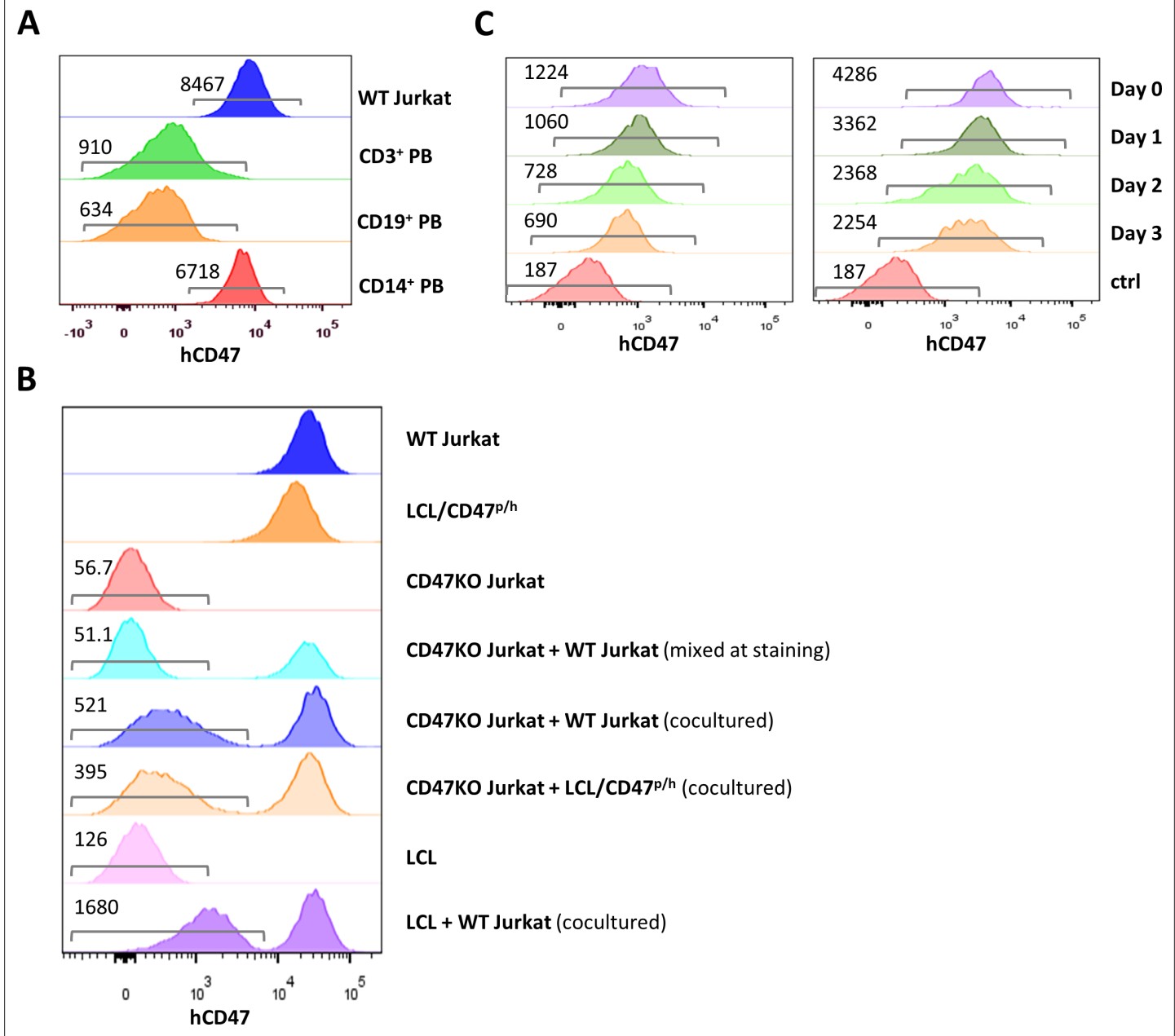

**Figure 2.** CD47 expression and CD47 cross-dressing on human T cell leukemia Jurkat cells. (**A**) CD47 expression on Jurkat cells and normal human CD3+, CD19+, and CD14+ peripheral blood cells (the numbers indicate median fluorescent intensity [MFI] of human CD47 staining). (**B**) CD47 expression on wild-type (WT) Jurkat cells, pig lymphoma cell line (LCL)/CD47p/h cells, CD47KO Jurkat cells, CD47KO cells mixed with WT Jurkat cells (mixed at the time of staining), CD47KO Jurkat cells cocultured (24 hr) with WT Jurkat or pig LCL/CD47p/h cells, pig LCL cells, and LCL cells cocultured (24 hr) with WT Jurkat cells. (**C**) PKH67-labeled pig LCL cells were cocultured with PHK26-labeled LCL/CD47p/h (left) or WT Jurkat (right) cells for 1 day, PHK67-labeled pig LCL cells are sorted and treated with mitomycin C (2 µg/ml) for 30 min (to stop cell division), then analyzed for hCD47 staining on pig LCL cells immediately (Day 0) and at the indicated times after cultured in media. The numbers in the figure indicate MFI of CD47 staining on gated cells.

The online version of this article includes the following figure supplement(s) for figure 2:

**Figure supplement 1.** Generation of CD47-deficient Jurkat cells.

**Figure supplement 2.** Multiple membrane proteins are cross-dressed during cell coculture.

**Figure supplement 3.** A functional cytoskeleton is not absolutely required for but may facilitate hCD47 cross-dressing.

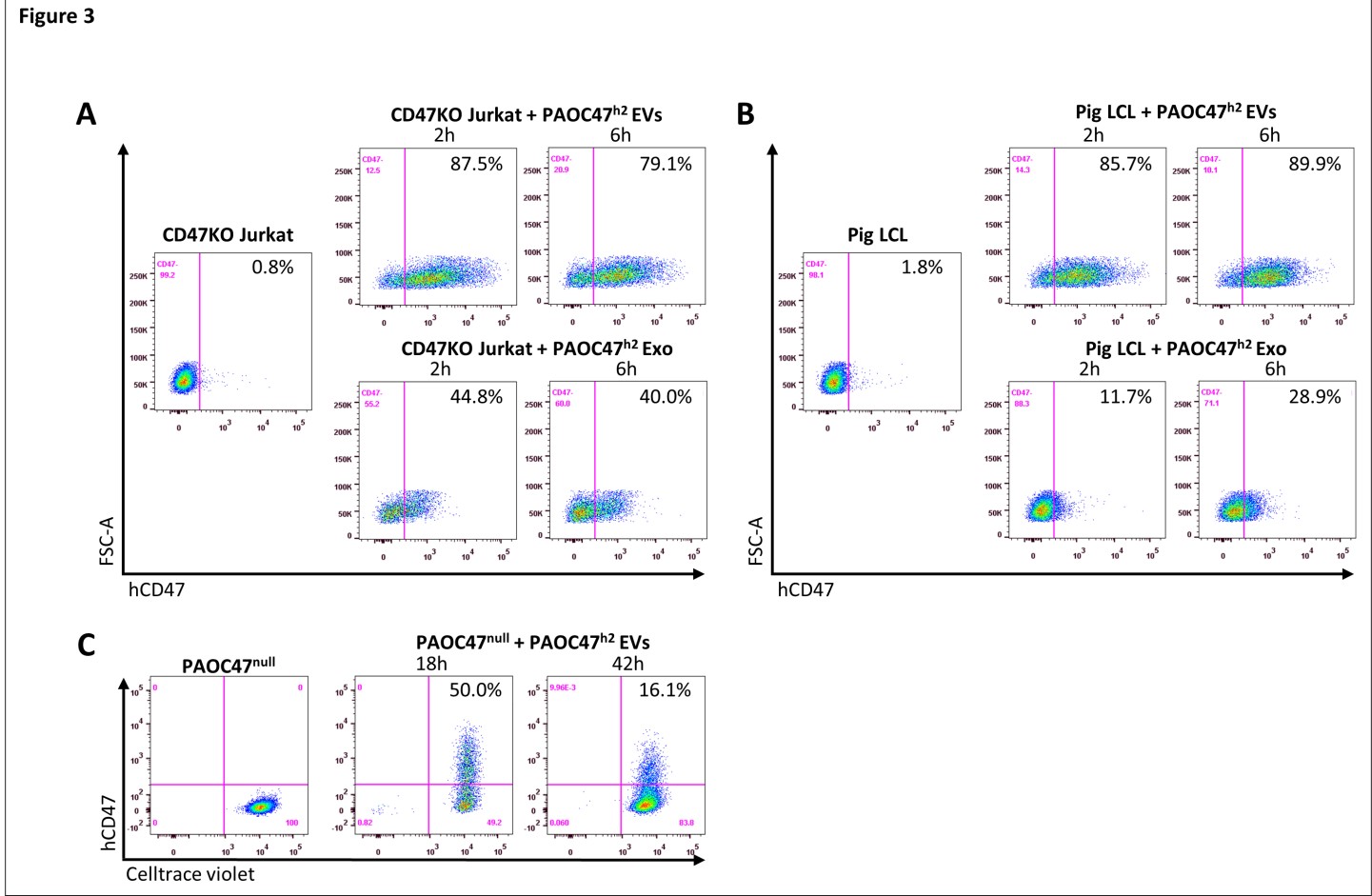

**Figure 3.** CD47 cross-dressing by extracellular vesicles (EVs) and exosomes (Exos). (**A–B**) CD47KO Jurkat cells (**A**) or pig lymphoma cell line (LCL) cells (**B**) were cultured in the absence (*left*) or presence (*right*) of EVs (*top*) or Exos (*bottom*) prepared from PAOC/CD47$^{h2}$ cell culture supernatants for 2 hr or 6 hr, and analyzed for hCD47 cross-dressing by flow cytometry using anti-hCD47-BV786 mAb. Representative flow cytometry profiles of three independent experiments were shown. (**C**) Celltrace violet-labeled PAOC/CD47$^{null}$ cells were cultured in the absence (*left*) or presence (*right*) of EVs prepared from PAOC/CD47$^{h2}$ cells for 18 hr or 42 hr, and analyzed for hCD47 cross-dressing by flow cytometry using anti-hCD47-BV786 mAb.

The online version of this article includes the following figure supplement(s) for figure 3:

**Figure supplement 1.** Rapid CD47 cross-dressing on pig lymphoma cell line (LCL) cells incubated with CD47-expressing extracellular vesicles (EVs).

**Figure supplement 2.** Cross-dressing involves membrane fusion.

## CD47 cross-dressing by extracellular vesicles and exosomes

We next investigated whether CD47 cross-dressing can be induced by EVs or requires direct cell-cell interaction. We analyzed hCD47 cross-dressing on CD47KO Jurkat cells, pig LCL cells, and PAOC/CD47$^{null}$ cells in the absence or presence of EVs prepared from PAOC/CD47$^{h2}$ cells. Flow cytometry analysis showed that CD47 cross-dressing occurred in both CD47KO human T-cell leukemia Jurkat cells (*Figure 3A*) and pig B-lymphoma LCL cells (*Figure 3B*) after incubation for 2 or 6 hr with PAOC47$^{h2}$ cell-derived EVs. To a less extent, both CD47KO Jurkat and LCL cells were also positively stained by anti-hCD47 after incubation with Exos released by PAOC47$^{h2}$ cells (*Figure 3A and B*). Similarly, hCD47 cross-dressing was detected in PAOC$^{null}$ cells after incubation with EVs from PAOC47$^{h2}$ cells (*Figure 3C*). In this experiment, PAOC$^{null}$ cells were labeled with florescence Celltrace violet prior to incubation with EVs to ensure there was no contamination by PAOC$^{h2}$ cells in the prepared EVs. After incubation with PAOC47$^{h2}$ EVs, a significant proportion of violet-labeled PAOC$^{null}$ cells became positive for hCD47 (*Figure 3C*). Of note, the frequency of hCD47$^{+}$ PAOC$^{null}$ cells at 42 hr was lower than that at 18 hr, which is most likely due to PAOC$^{null}$ cell proliferation and EV exhaustion/degradation. Pig LCL cells were also stained positive for hCD47 after incubated with EVs from hCD47-tg pig LCL cells

or WT Jurkat cells, in which hCD47 cross-dressing was detected 1 hr after incubation with EVs and the levels remained stable or increased at 24 hr (*Figure 3—figure supplement 1*). These results indicate that CD47 cross-dressing could be induced independently of cell-cell contact by EVs, including Exos. Again, a functional cytoskeleton is not absolutely required for hCD47 cross-dressing, as hCD47 was clearly detected on cytochalasin D-pretreated LCL cells after incubation with EVs from hCD47-tg LCL cells, though the level was moderately reduced (*Figure 2—figure supplement 3B*). Furthermore, cross-dressing was not diminished by treatment with trypsin/EDTA, indicating that membrane fusion is involved in EV cross-dressing (*Figure 3—figure supplement 2*).

## Protection against phagocytosis by cross-dressed CD47

We next determined whether cross-dressed CD47 can act as a marker of self to protect the cells against phagocytosis. We first investigated the binding potential of cross-dressed hCD47 with human SIRPα. CD47KO Jurkat cells were cocultured without or with EVs from PAOC/CD47[h2] cells for 5 hr, washed, and incubated with recombinant human SIRPα-Fc chimera for 1 hr. Binding of human SIRPα fusion protein to CD47KO Jurkat cells was then measured using fluorochrome-labeled anti-human IgG Fc antibody. Flow cytometry analysis showed that human SIRPα fusion protein was able to bind CD47KO Jurkat cells cultured with EVs but not those cultured without EVs (*Figure 4A*). The data indicate that cross-dressed hCD47 on CD47KO Jurkat cells can bind human SIRPα.

We then performed phagocytic assay to determine the potential of cross-dressed hCD47 to protect pig cells or CD47KO human leukemia cells against phagocytosis by human monocyte-derived macrophages. PAOC47[null] and PAOC/CD47[h2] cells were cultured for 48 hr, then hCD47 cross-dressed PAOC47[null] cells were sorted out (*Figure 4B*) and their susceptibility to phagocytosis by human macrophages was determined in comparison to PAOC47[null] and PAOC/CD47[h2] cells that were cultured separately (*Figure 4C*). As expected, PAOC47[null] cells were significantly more sensitive than PAOC/CD47[h2] cells to phagocytosis (*Figure 4C*). However, hCD47 cross-dressing effectively reduced the susceptibility of PAOC47[null] cells to phagocytosis by human macrophages, to a level comparable to that of PAOC/CD47[h2] cells (*Figure 4C*). Human CD47 cross-dressing protects not only xenogeneic pig cells, but also human leukemia cells, against phagocytosis by human macrophages. In phagocytic assays where CD47KO Jurkat cells showed significantly greater phagocytosis than WT Jurkat cells, pre-incubation of CD47KO Jurkat cells with EVs released by PAOC/CD47[h2] cells was found highly effective in reducing their phagocytosis by human macrophages (*Figure 4D*). These results indicate that human CD47 cross-dressing can act as a functional ligand for human SIRPα and deliver 'don't eat me' signals to human macrophages.

## Ligation of autogenous but not cross-dressed CD47 induces death in Jurkat cells

Ligation of cell surface CD47 by its ligand thrombospondin-1 (TSP-1) (*Saumet et al., 2005*), CD47-binding peptides of TSP-1 (*Martinez-Torres et al., 2015*) or CD47 antibodies (*Mateo et al., 1999*) has been shown to induce death in varying types of cells. In line with these reports, we observed that human SIRPα-Fc fusion proteins could induce cell death in a dose-dependent manner in WT, but not CD47KO, human T-cell leukemia Jurkat cells (*Figure 5—figure supplement 1*; *Figure 5A and B*). The cell death observed in WT Jurkat cells was induced by CD47 ligation with hSIRPα-Fc proteins, as cell death was minimally detectable in WT Jurkat cells that were cultured simultaneously without hSIRPα-Fc proteins. To determine whether cross-dressed CD47 on Jurkat cells may also induce cell death, we compared the susceptibility to cell death induced by SIRPα-Fc proteins among WT, CD47KO, and hCD47 cross-dressed CD47KO Jurkat cells. CD47 cross-dressing was performed on GFP[+] CD47KO Jurkat cells by incubation for 2 hr with PAOC/CD47[h2] EVs. To induce cell death, GFP[+] CD47KO or hCD47 cross-dressed GFP[+] CD47KO Jurkat cells were cocultured, respectively, with an equal number of control WT Jurkat cells ($5 \times 10^4$ each) in the presence of human SIRPα-Fc for 1 hr. The cocultured cells were then stained with anti-hCD47 (BV786) mAb, and cell death in WT (hCD47[+]GFP[-]), CD47KO (CD47[-]GFP[+]), and hCD47 cross-dressed CD47KO (CD47[low]GFP[+]) Jurkat cells were measured. Of note, binding of hSIRPα-Fc proteins to cell surface CD47 (either native or cross-dressed) could partially block subsequent staining with anti-hCD47 antibodies and thus, the cells cultured with hSIRPα-Fc showed relatively lower hCD47 staining than those cultured without (*Figure 5—figure supplement 2*, *Figure 5*). We found that SIRPα-Fc proteins induced significant cell

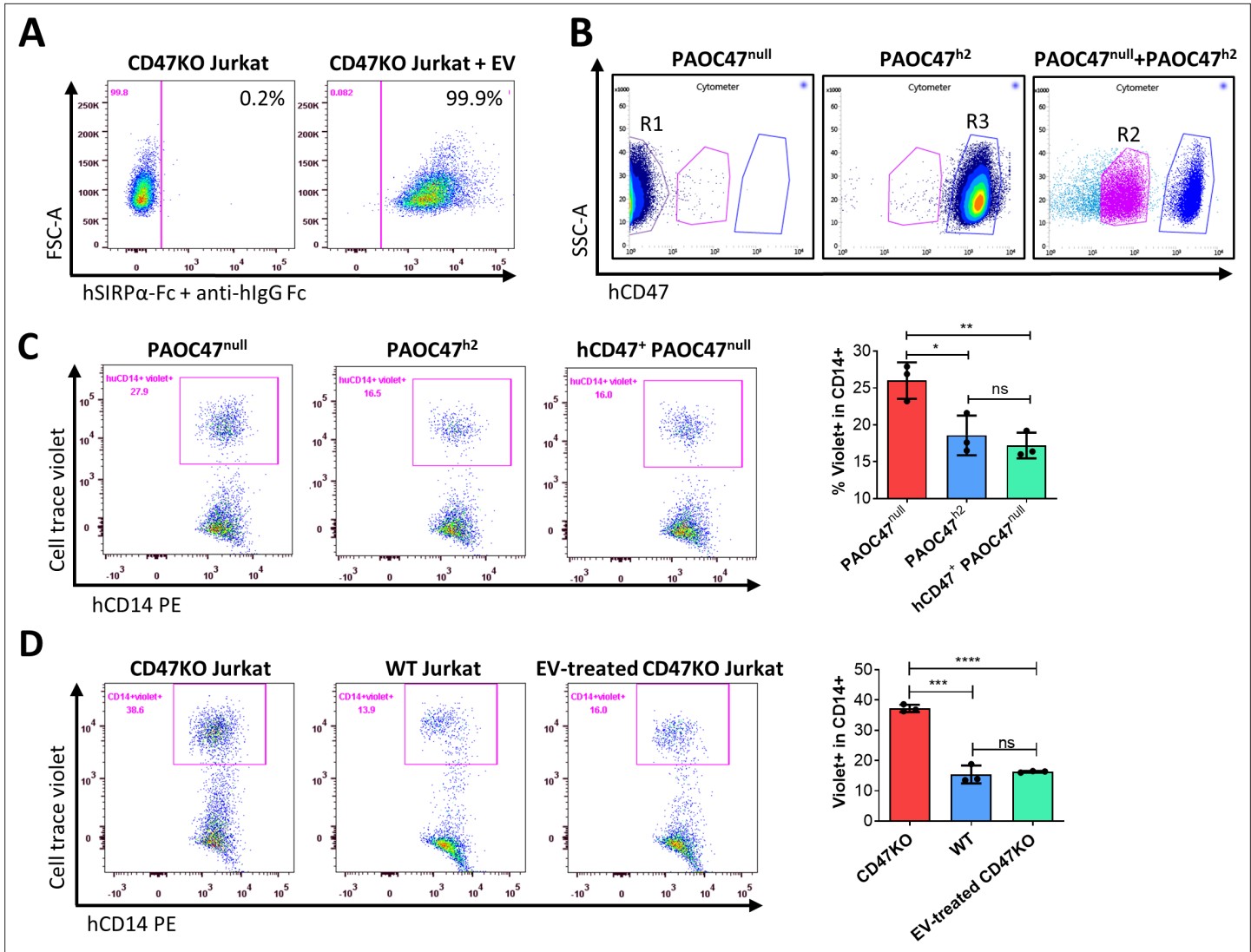

**Figure 4.** Protection against phagocytosis by cross-dressed CD47. (**A**) CD47KO Jurkat cells were cocultured without (*left*) or with (*right*) extracellular vesicles (EVs) from PAOC/CD47h2 cells at 37°C for 5 hr, then washed and incubated with recombinant human SIRPα-Fc chimera at 37°C for 1 hr. The binding of SIRPα-Fc proteins to CD47KO Jurkat cells was visualized by staining with APC-conjugated mouse anti-human IgG Fc mAb. Representative flow cytometry profiles of two independent experiments are shown. (**B**) PAOC47null and PAOC47h2 cells were cultured alone (*left* and *middle*) or together (*right*) for 48 hr and stained using anti-hCD47-BV786 mAb, then PAOC47null (**R1**), PAOC47h2 (**R3**), and hCD47+ (i.e. hCD47 cross-dressed) PAOC47null (R2) cells sorted from cocultures were used immediately for phagocytic assay. (**C**) PAOC47null (**R1**), PAOC47h2 (**R3**), or sorted hCD47 cross-dressed PAOC47null (R2) cells were labeled with Celltrace violet and incubated with human macrophages for 2 hr, then phagocytosis was determined by flow cytometry using anti-human CD14 mAb. Shown are representative flow cytometry profiles (left) and levels (right; mean ± SDs; n=3) of phagocytosis (i.e. percentages of human macrophages that have engulfed violet + target cells (CD14+violet+) in human CD14+ macrophages). Representative results of three independent experiments are shown. (**D**) CD47KO Jurkat, wild-type (WT) Jurkat, and CD47KO Jurkat cells pre-incubated with EVs from PAOC/CD47h2 cells were labeled by Celltrace violet and cocultured with human macrophages for 2 hr, then phagocytosis was analyzed by flow cytometry. Shown are representative flow cytometry profiles (left) and levels (right; mean ± SDs; n=3) of phagocytosis (i.e. percentages of hCD14+violet+ in total hCD14+ macrophages). Representative results of two independent experiments are shown. *, p<0.05; **, p<0.01; ***, p<0.001; ****, p<0.0001; ns, not significant (two-tailed unpaired t-test).

death in WT Jurkat cells regardless of whether they were cocultured with CD47KO (*Figure 5C and D*) or with hCD47 cross-dressed CD47KO (*Figure 5E and F*). CD47 cross-dressing did not increase the sensitivity to cell death induced by SIRPα-Fc proteins, and cell death was minimally detectable in both CD47KO (*Figure 5C and D*) and hCD47 cross-dressed CD47KO (*Figure 5E and F*) Jurkat cells. However, CD47KO Jurkat cells regained the sensitivity to apoptosis induced by CD47 agonists after being transfected to express CD47 (*Figure 5—figure supplement 3*). These results indicate that,

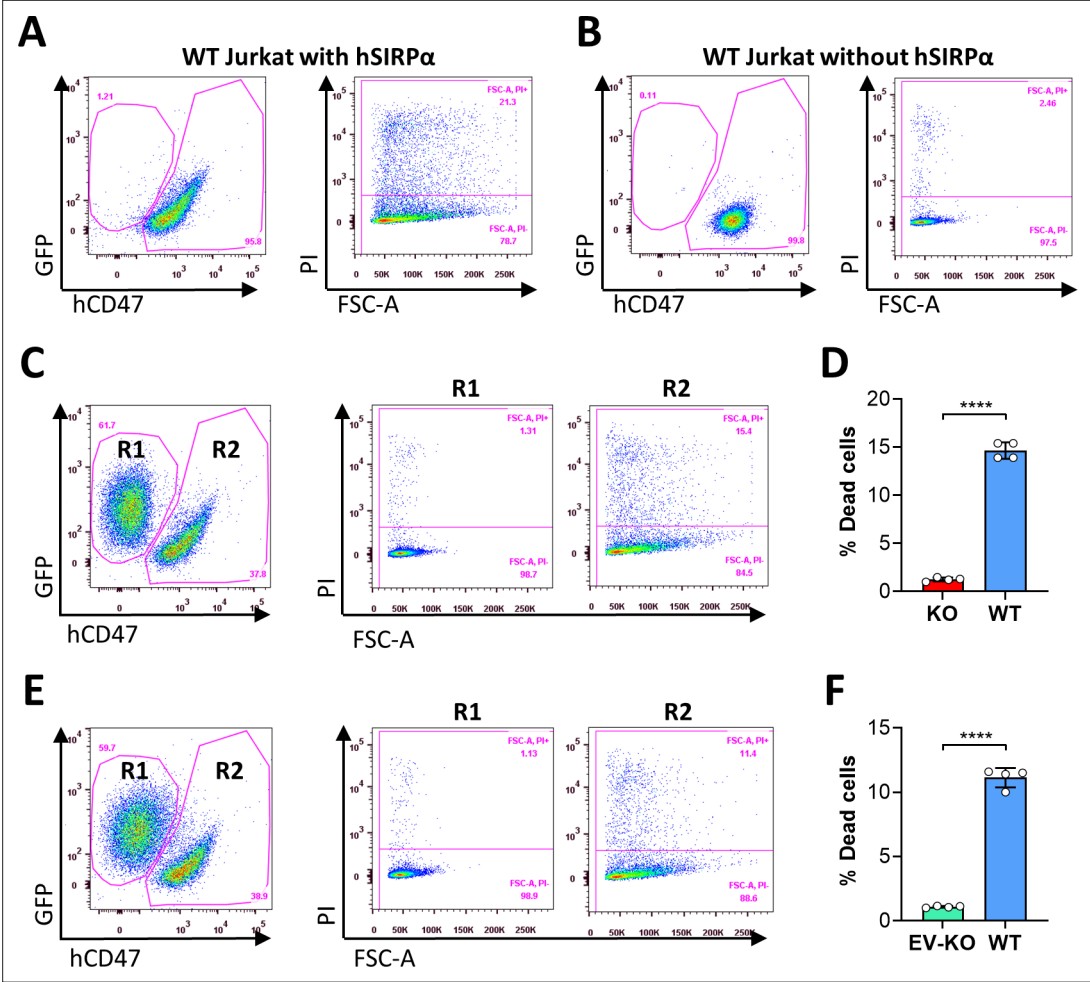

**Figure 5.** Ligation of autogenous but not cross-dressed CD47 induces cell death. (**A–B**) 5×10⁴ wild-type (WT) Jurkat cells (GFP⁻) were incubated in the presence (**A**) or absence (**B**) of 2.5 µg/ml hSIRPα-Fc proteins at 37°C for 1 hr and stained with anti-hCD47 mAb and Propidium Iodide (PI). Representative flow cytometry profiles show dead cells (PI⁺) in hCD47⁺ WT Jurkat population. (**C–D**) CD47KO Jurkat (GFP⁺) and WT Jurkat cells were mixed (at 1:1 ratio; 5×10⁴ each) and cocultured in the presence of 2.5 µg/ml human SIRPα-Fc for 1 hr, then stained with anti-hCD47 mAb and PI. Shown are representative flow cytometry profiles (**C**) and percentages (D; mean ± SDs) of PI⁺ dead cells in gated CD47KO (R1, CD47⁻GFP⁺) and WT (R2, hCD47⁺GFP⁻) Jurkat cells. (**E–F**) CD47KO Jurkat (GFP⁺) cells were incubated for 2 hr with PAOC/CD47ʰ² extracellular vesicles (EVs), then mixed with WT Jurkat cells (at 1:1 ratio; 5×10⁴ each) and cocultured in the presence of 2.5 µg/ml human SIRPα-Fc for 1 hr. The cells were stained with anti-hCD47 mAb and PI. Shown are representative flow cytometry profiles (**E**) and percentages (F; mean ± SDs) of PI⁺ dead cells in gated EV-treated CD47KO (R1, CD47ˡᵒʷGFP⁺) and WT (R2, hCD47⁺GFP⁻) Jurkat cells. ****, p<0.0001 (two-tailed unpaired t-test). Of note, the cells cultured with hSIRPα-Fc proteins (**A, C and E**) showed reduced hCD47 staining (as detailed in *Figure 5—figure supplement 2*). Results shown are representative of three independent experiments.

The online version of this article includes the following figure supplement(s) for figure 5:

**Figure supplement 1.** SIRPα-Fc-induced cell death in wild-type (WT) and CD47KO Jurkat cells.

**Figure supplement 2.** Prior incubation with hSIRPα-Fc proteins partially blocks subsequent staining by anti-hCD47 mAb.

**Figure supplement 3.** CD47-transfected CD47KO Jurkat cells are sensitive to apoptosis induced by CD47 agonists.

unlike autogenous CD47, cross-dressed CD47 on Jurkat cells does not induce cell death upon ligation with SIRPα-Fc proteins.

## Discussion

Given its strong inhibitory effects on macrophage activation and phagocytosis, transgenically overexpressed CD47 is considered an effective means of preventing transplant rejection. A major obstacle impeding the translation of xenotransplantation into clinical therapies is vigorous xenograft rejection (*Yang and Sykes, 2007*), and the lack of functional interaction in CD47-SIRPα pathway is a key

mechanism triggering macrophage xenoimmune responses (*Yang, 2010*). Studies have shown that the use of gene-edited pigs carrying human CD47 is effective in protecting against xenograft rejection by macrophages in non-human primates (*Ide et al., 2007*; *Tena et al., 2017*; *Watanabe et al., 2020*). Recently, transgenic overexpression of CD47 was also successfully used in combination with other approaches, such as deletion of HLA molecules, to generate hypoimmunogenic pluripotent stem cells (*Han et al., 2019*; *Deuse et al., 2019*). Here we found that EVs and Exos released from pig cells transgenically overexpressing hCD47 can mediate hCD47 cross-dressing on surrounding pig cells. Furthermore, hCD47 cross-dressed on pig cells can interact with human SIRPα and inhibit phagocytosis by human macrophages. Such CD47 cross-dressing occurs not only in the pig-to-pig combination, but also in the human-to-human, pig-to-human, and human-to-pig directions. These results provide a new mechanism for the inhibition of phagocytosis by the approach of transgenically expressing CD47.

In both xenogeneic and allogeneic settings, a high level of transgenic CD47 expression was found to be essential for its immune inhibitory effect. CD47 is not only a ligand of SIRPα, but also a signaling receptor that mediates a variety of functions, including apoptosis, cell cycle arrest, and senescence. It was reported that deletion of CD47 improves survival, proliferation, and function of endothelial cells, leading to increased angiogenesis and neovascularization both in vitro and in vivo (*Meijles et al., 2017*; *Gao et al., 2016*; *Gao et al., 2017*). In line with these observations, CD47 deletion from organ grafts was reported to ameliorate renal ischemia/reperfusion injury (*Isenberg and Roberts, 2019*) and cardiac allograft rejection (*Chen et al., 2019*). Using a pig-to-baboon kidney xenotransplantation model, a recent study suggested that widespread expression of hCD47 in the pig kidney was associated with increased vascular permeability and systemic edema, presumably due to upregulated TSP-1, and hence, TSP-1-CD47 signaling in the graft (*Takeuchi et al., 2021*). Our study showed that, although CD47 cross-dressed on cells may bind CD47 ligands, the ligand engagement does not induce CD47 signaling to cause apoptosis. We found that ligation of hSIRPα-Fc proteins with autogenous, but not cross-dressed, CD47 on Jurkat cells induces cell death. This study suggests that using a pig vascularized organ with hCD47 overexpression in some cells, which are more sensitive to macrophage attack but relatively resistant to CD47 signaling-induced deleterious effects, may improve the outcomes of xenotransplantation.

Emerging evidence indicates that CD47-SIRPα signaling plays an important role in regulating DC activation, and hence, T cell priming. The initial evidence for a role of CD47 in controlling DC activation was obtained in a mouse model of DST, in which DST using WT cells induces donor-specific tolerance, but DST using CD47-deficient cells paradoxically induces SIRPα$^+$ DC activation and augments anti-donor T cell responses (*Wang et al., 2010*). A similar finding was made in a mouse model of hepatocyte allotransplantation where WT hepatocytes promote allograft survival, but CD47KO hepatocytes exacerbated rejection (*Zhang et al., 2016*). Although it was not tested directly, it is conceivable that CD47 cross-dressed on cells, which induces sufficient SIRPα signaling to inhibit macrophages, may suppress SIRPα$^+$ DC activation. Thus, in addition to inhibition of phagocytosis, CD47 cross-dressing may also attenuate anti-donor T cell responses when transplants are performed using CD47-overexpressing donors.

CD47-SIRPα signaling is an important component of the tumor immune microenvironment. CD47 upregulation was found in many types of cancer cells, in which CD47 provides an important mechanism to evade macrophage killing (*Jaiswal et al., 2009*; *Chan et al., 2009*). CD47 upregulation in tumors also significantly contributes to tumor-induced T cell suppression, as the antitumor activity of CD47 blocking treatment is associated with CD11c$^+$ DC activation and is largely T cell-dependent (*Liu et al., 2015b*, *Chen et al., 2020*; *Li et al., 2020*). In the present study, CD47 cross-dressing was found in human T-cell leukemia Jurkat and pig B-lymphoma LCL cells, suggesting a possible involvement of CD47 cross-dressing in the formation of a tumor immunosuppressive microenvironment. In addition, CD47 on EVs and CD47 cross-dressed on tumor cells may also neutralize CD47 ligands, such as TSP-1 that has been shown to induce CD47 activation leading to apoptosis in tumor and endothelial cells (*Martinez-Torres et al., 2015*), hence, favoring tumor growth.

While the mechanisms of EV-mediated exchange of biological information and materials between cells remain poorly understood (*Raposo and Stahl, 2019*), EV-induced antigen cross-dressing has been reported to play an important role in the regulation of immune responses, including alloantigen recognition and allograft rejection (*Zeng and Morelli, 2018*; *Gonzalez-Nolasco et al., 2018*). Earlier studies have shown that CD47, as a 'don't eat me' signal, is essential for EVs to elicit biological

function by preventing their clearance by macrophages (*Kamerkar et al., 2017*). Here we report that EV-induced CD47 cross-dressing possesses partial activity of the autogenous CD47, such as the ability to initiate inhibitory SIRPα signaling, and therefore offers means of separating the desired and harmful effects of CD47.

# Materials and methods

## Cell culture

Jurkat (J.RT3-T3.5) cell line was purchased from ATCC (named WT Jurkat to distinguish from CD47KO Jurkat cells) and the identity has been authenticated by STR profiling provided by China Center for Type Culture Collection (CCTCC). Jurkat cells were grown in Dulbecco's Modified Eagle's Medium supplemented with 10% Fetal Bovine Serum (FBS) (Atlanta Biologicals), GlutaMax and 100 U/ml penicillin and streptomycin. Pig LCL cell is a pig B cell lymphoma cell line (LCL-13271) derived from Swine Leukocyte Antigen AD (SLA$^{AD}$) miniature swine with post-transplantation lymphoproliferative disease (PTLD), which is kindly provided by Christene Huang (Harvard Medical School). The markers used for characterization of pig LCL cells are CD2, CD25 and anti-mu heavy chain (details were published previously) (*Cho et al., 2007*). Human CD47 (hCD47) transgenic (tg) pig B-LCL cell line (hCD47-tg LCL) and control pig LCL cell line were generated by transfecting LCL cells with pKS336-hCD47 or empty pKS336 vector, respectively, as described previously (*Ide et al., 2007*). Pig LCL cells were grown in RPMI 1640 Medium supplemented with 10% FBS, GlutaMax, MEM Non-Essential Amino Acids Solution, sodium pyruvate, 5 μM 2-mercaptoenthaol and 100 U/ml penicillin and streptomycin. Human CD47-tg porcine BMCs were harvested from SLA-defined miniature swine with hCD47 transgene (*Watanabe et al., 2020*), and BMCs were grown in Dulbecco's Modified Eagle's Medium supplemented with 10% FBS, GlutaMax and 100 U/ml penicillin and streptomycin. Pig aortic endothelial cells (PAOC) immortalized with SV40 were purchased from ABM (catalog # T0448), and the endothelial cell characterization was determined by expression of VE cadherin and CD31, and uptake of fluorescent DiI-Ac-LDL (measured by DiI-Ac-LDL Kit; Cell Applications, Cat # 022K). All PAOC cell lines were grown in Porcine Endothelial Cell Growth Medium (Cell Applications, Cat # P211-500) supplemented with 10% FBS. All cell lines were confirmed negative for mycoplasma contamination by MycoStrip mycoplasma detection kit (InvivoGen, Cat# rep-mys-10). Trypsin/EDTA was used for cell dissociation. Media and reagents for cell culture were purchased from GIBCO. Where indicated, cells were treated with mitomycin C (2 μg/mL for 30 min; Sigma-Aldrich) or cytochalasin D (2 μg/mL for 60 min; Thermo Fisher).

## Generation of CD47KO Jurkat and PAOC sublines

CRISPR small guide RNA (sgRNA) for disrupting hCD47 in Jurkat cells was designed using the online tools (https://crispr.mit.edu), with sequence targeting the exon 2 of hCD47 (CTACTGAAGTATACGT AAAG-TGG [PAM]). The sgRNA was cloned into the pL-CRISPR.EFS.GFP lentiviral vector which was a gift from Benjamin Ebert (Addgene plasmid no. 57818) for co-expression with Cas9 (*Heckl et al., 2014*). Lentiviral particles were produced by co-transfection of a three-plasmid system consisting of the pL-CRISPR.EFS. GFP vector and packaging plasmids (pVSV-G and pΔ) using CaCl$_2$ into 293T cells in 175 cm$^2$ flasks. Lentivirus supernatant was collected 48 hr post-transfection, concentrated by ultracentrifugation at 22,000 rpm for 2.5 hr (Beckman Coulter, Optima XE-90) and stored at –80°C until use. GFP$^+$ cells were sorted 3 days after lentivirus transduction, then sorted GFP$^+$ cells were expanded and assessed for CD47 expression by staining with BV786-conjugated anti-hCD47 mAb B6H12 (BD Bioscience) and PE-conjugated anti-hCD47 mAb CC2C6 (Biolegend). CD47-negative Jurkat cells were established by four rounds of cell sorting. We also generated Jurkat cells that express only transgenic hCD47 by transduction of CD47-negative Jurkat cells with pLVX lentiviral vector encoding hCD47 isoform 2 cDNA.

PAOC cells were immortalized with Lenti-hTERT virus (ABM; cat no. G200) following manufacturer's instructions and clonal sorting/expansion. Alpha GAL KO pAOC-SV40-hTERT (GTKO) cell line (PAOC/CD47$^P$) was created via nucleofection of pAOC-SV40-hTERT with plasmid expressing Cas9 protein (GeneART CRISPR Nuclease Vector, Invitrogen) and guide RNA targeting GGTA-1 gene (aGal protein, guide RNA sequence used: TCATGGTGGATGATATCTCC) using Lonza 4D-Nucleofector, followed by clonal sorting/expansion. PAOC/CD47$^{null}$ was created via nucleofection of PAOC/CD47$^P$ cells with

plasmid expressing Cas9 protein and guide RNA targeting the pig CD47 gene (guide RNA sequence used: TCACCATCAGAATTACTACA) using Lonza 4D-Nucleofector. PAOC/CD47$^{null}$ and PAOC/CD47$^p$ cell lines expressing hCD47 short (305aa, 16aa short intracellular domain, NM_198793, NP_942088, PAOC/CD47$^{h2}$, and PAOC/CD47$^{p/h2}$) or hCD47long isoforms (323aa, 34aa long intracellular domain, NM_001777, NP_001768, PAOC/CD47$^{h4}$, and PAOC/CD47$^{p/h4}$) (*Reinhold et al., 1995*) were created via nucleofection with plasmid expressing Cas9 protein (GeneART CRISPR Nuclease Vector, Invitrogen) and guide RNA targeting AAVS1 safe harbor site using Lonza 4D-Nucleofector, followed by clonal sorting/expansion.

## Flow cytometric analysis

CD47 expression or cross-dressing on cells was determined by direct staining with BV786- or AF647-conjugated anti human specific CD47 mAb B6H12 (abbreviated anti-hCD47; BD Bioscience) or PE-conjugated anti-human CD47 mAb CC2C6 (with cross-reactivity to pig CD47 and thus, referred to as anti-h/pCD47; Biolegend). The level of cell surface CD47 is expressed as median fluorescent intensity. FITC(fluorescein isothiocyanate)-conjugated anti-pig SIRPα mAb (clone BL1H7) was from Abcam; recombinant human SIRPα/CD172a Fc chimera protein, CF (cat no. 4546-SA-050) was from R&D system; APC-conjugated anti-human IgG Fc (clone HP6017), PE-conjugated anti-human CD90 (clone 5E10), AF647-conjugated anti-human HLA-ABC (clone W6/32) were from Biolegend; PKH26 and PKH67 were from Sigma-Aldrich. For analysis of CD47 expression on human PBMCs, single-cell suspensions were incubated with anti-hCD47 mAb B6H12 in combination with fluorochrome-conjugated anti-human CD45 (clone HI30), CD19 (clone HIB19), CD3 (clone SK7), and CD14 (clone M5E2; all from Biolegend). Dead cells were identified by staining with propidium iodide or 7-AAD. All samples were collected on Flow Cytometer (Fortessa and Celesta, Becton Dickinson), and data were analyzed by Flowjo software (Tree Star).

## Purification of extracellular vesicles

EVs and Exos from cell culture supernatants were purified by a standard differential centrifugation protocol as previously reported (*Chen et al., 2018*). In brief, bovine Exos were depleted from FBS by overnight centrifugation at 100,000 g, PAOC/CD47$^{h2}$ cells were cultured in media supplemented with 10% Exos-depleted FBS for EVs and Exos purification from cell culture supernatants. Supernatants collected from 48-hr cell cultures were centrifuged at 2000 g (3000 rpm) for 20 min to remove cell debris and dead cells. EVs were pelleted after centrifugation at 16,500 g (9800 rpm) for 45 min (Beckman Coulter, Optima XE-90) and resuspended in PBS. The pelleted Exos from above supernatants were further centrifuged at 100,000 g (26,450 rpm) for 2 hr at 4°C (Beckman Coulter, Optima XE-90) and resuspended in PBS. EVs and Exos from total $1.3\times10^7$ PAOC/CD47$^{h2}$ cells cultured for 48 hr were concentrated in 250 ul and 400 ul PBS, respectively, 10 ul of each was used for CD47 cross-dressing.

## Preparation of human macrophages

Blood from healthy volunteers was used to prepare peripheral blood mononuclear cells (PBMCs) by density gradient centrifugation. PBMCs were added at $3\times10^6$ per well in a 24-well plate, and unattached cells were removed from the plate on the second day. Attached cells were then differentiated to macrophages by 8–9 days of culture in Iscove's Modified Dulbecco's Medium (IMDM) (Gibco) + GlutaMax (Thermo fisher scientific) supplemented with 10% AB human serum (Gemini Bio-products, Inc), containing 10 ng/ml human M-CSF (PeproTech) and 100 U/ml penicillin and streptomycin (Gibco). The use of human blood samples was approved by the Institutional Review Board of Columbia University Medical Center.

## Flow cytometry-based phagocytic assay

Macrophages generated as above were harvested from plates using Trypsin-EDTA (Thermo fisher scientific). The indicated target cells were labeled with Celltrace violet (Thermo fisher scientific) according to the manufacturer's protocol, and phagocytic assay was performed by co-culturing $6\times10^4$ Celltrace Violet-labeled target cells with $3\times10^4$ human macrophages for 2 hr in ultra-low attachment 96-well flat bottom plates in IMDM + GlutaMax without antibiotics or serum added. All cells were harvested after coculture, and phagocytosis was determined by flow cytometry analyses, in which the

phagocytic ratio is calculated as the percentage of macrophages that engulfed target cells (human $CD45^+CD14^+Celltrace violet^+$) among total macrophages (human $CD45^+CD14^+$).

## Statistical analysis

Data were analyzed using GraphPad Prism (version 8; San Diego, CA) and presented as mean value ± SDs. The level of significant differences in group means was assessed by student's t-test, and a p-value of ≤0.05 was considered significant in all analyses herein.

## Acknowledgements

This work was supported by NIH PO1 AI045897 and a fund from The First Hospital of Jilin University. The CCTI Flow Cytometry Core used was funded in part through an NIH Shared Instrumentation Grant (1S10RR027050).

## Additional information

### Funding

| Funder | Grant reference number | Author |
|---|---|---|
| National Institutes of Health | AI045897 | Megan Sykes |
| The First Hospital of Jilin University | | Yong-Guang Yang |
| NIH | PO1 AI045897 | Megan Sykes |
| NIH Shared Instrumentation | 1S10RR027050 | Megan Sykes |

The funders had no role in study design, data collection and interpretation, or the decision to submit the work for publication.

### Author contributions

Yang Li, Conceptualization, Data curation, Formal analysis, Investigation, Methodology, Writing – original draft, Writing – review and editing; Yan Wu, Xiaodan Wang, Andre Dharmawan, Hui Wang, Data curation, Investigation, Writing – review and editing; Elena A Federzoni, Data curation, Formal analysis, Validation, Investigation, Writing – review and editing; Xiaoyi Hu, Data curation, Formal analysis, Investigation, Writing – review and editing; Robert J Hawley, Sean Stevens, Resources, Supervision, Funding acquisition, Writing – review and editing; Megan Sykes, Conceptualization, Resources, Supervision, Funding acquisition, Writing – review and editing; Yong-Guang Yang, Conceptualization, Data curation, Formal analysis, Supervision, Funding acquisition, Investigation, Writing – original draft, Project administration, Writing – review and editing

### Author ORCIDs

Yang Li  http://orcid.org/0000-0003-4623-4489
Yong-Guang Yang  http://orcid.org/0000-0003-4985-1247

### Decision letter and Author response

Decision letter https://doi.org/10.7554/eLife.73677.sa1
Author response https://doi.org/10.7554/eLife.73677.sa2

## Additional files

### Supplementary files
- MDAR checklist
- Source data 1. CD47 crossdressing source data E-life R1.

## Data availability

Figure 1, 4, 5; Figure 2-figure supplement 2, Figure 2-figure supplement 3, Figure 3-figure supplement 1, Figure 3-figure supplement 2, Figure 5-figure supplement 1, Figure 5-figure supplement 3 - Source Data contain the numerical data used to generate the figures.

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
