## [Editor Report]

This study by Li et al., describes an important molecular mechanism that limits the rejection of xenotransplant or allogenic tissue. By presenting compelling evidence, that describes a mechanism to cross-dress xenogeneic cells with human CD47. Li et al., investigated if this process can improve grafted tissue survival by halting phagocytosis of xenogeneic cells by host human monocytes while also diminishing SIRP-a mediated apoptosis. This work will be of interest to basic and transplant immunology.

---

## [Decision Letter]

**Decision letter after peer review:**

Thank you for submitting your article "CD47 cross-dressing by extracellular vesicles expressing CD47 inhibits phagocytosis without transmitting cell death signals" for consideration by *eLife*. Your article has been reviewed by 3 peer reviewers, and the evaluation has been overseen by a Reviewing Editor and Satyajit Rath as the Senior Editor. The following individuals involved in the review of your submission have agreed to reveal their identity: Dennis Discher (Reviewer #3).

Essential revisions:

1) Figure 1

The rationale for the use of CD47 isoforms 2 and 4 requires further detail a full explanation.

Please provide kinetics or the dynamic of CD47 acquisition in recipient cells.

Is cross-dressing blocked at 4deg?

Does cross-dressing require cell-to-cell contact?

Does cross-dressing require a functional cytoskeleton?

Do other membrane proteins get transferred as well or is it a peculiarity of CD47h2/h4 to be embedded in cross-dressing vesicles released from donor cells?

Does cross-dressed CD47 resist to EDTA washes (one might anticipate that tethered CD47 would not resist while CD47 actually transferred by membrane fusion would)?

What is the persistence of crossdressed CD47 on recipient cells (post-separation from donor cells)?

b) Why "ligation of the autogenous, but not cross-dressed, CD47 induced cell death." (ii) Please show the extracellular vesicles and/or exosomes, do or do not transfer hCD47 mRNA or even DNA (e.g. plasmid, even episomal). (iii) Determine the reason(s) and implications for function (e.g. phagocytosis) for the very broad distribution of intensities on cells that supposedly have 'all' received hCD47. (iv) Show whether "florescence Celltrace violet" transfers from labelled cells to unlabeled cells (Figure 1c). (v) Determine why results for h2 and h4 spliceforms are variable; sometimes the same (e.g. Figure 1a) versus sometimes different (Figure 1b).

c) In Figure 1A-B: How did the authors gate on the PAOCp in the coculture? Congenic marker, cell tracker? 1C is the right way to do it as surface markers/ membrane stains can't be relied upon. The histograms may be misleading as the shifted curves can include both the acceptor cells that capture CD47h and donor cells that had reduced CD47h. 1D: It's not possible to say that it's fully SIRP-a independent as the increase in the CD47 cross-dressing looks more modest compared to 1-A,B,C. However, this may be due to an intrinsic difference between the starting levels of CD47 on tg LCL and PAOC. Yet, this reviewer recommends changing the statement at line 131 as such: "…., indicating that hCD47 cross-dressing can occur in a SIRPα-independent manner."

2) Figure 2

In Figure 2 Please address labelling and gating the cells in the coculture (see comments 1c).

3) Figure 3

The titration of EVs/Exos should be performed to highlight the efficiency of the cross-dressing process.

The yield of EV/Exos production is explained in order to get an idea if the concentration/quantity of EVs correspond to a reasonable amount of cells and how this number compares to the number of cross-dressed "recipient" cells.

Does membrane acquisition upon EV/Exos involve membrane fusion?

(this can be addressed using fluorescent membrane probes).

In Figure 3C: Why did the authors cocultured cells for 18 and 42 hours? At 42h PAOC cells line would be expected to divide and dilute the CTV, why didn't they? Also, the fitness of the cells might be impaired at the late time points, have the authors performed a viability staining and gated out the dead cells?

4) Figure 4

It is unclear if CD14+CTV+ events all correspond to phagocytosis or doublets formation upon tethering of both cell types. Imaging experiments using confocal microscopy on sorted CD14+CTV+ events could easily quantify this. Please add quantitative imaging data.

This assay requires rigorous viability staining, a robust gating strategy to identify the donor and acceptor cells of the co-culture and controls with single cells. Without those, how do the authors claim that the CTV+ cells in the gate are monocytes that had phagocytosed the CTV-labeled cells?

4B: The population sorted is likely a mix of donor and acceptor cells and contains CD47 expressing cells too. This should have been carefully carried out by labeling the cells with cell trackers prior to culture and sorting them according to the label.

4C: What does the CD14 staining of PAOC cells before and after coculture look like? In systems where cross-dressing occurs, it's hard to rely on the surface markers as they have mobility between cells. Have the authors considered labelling donor and acceptor cells with two different cell trackers to set the gates properly and independent of surface markers?

5) Figure 5

SIRPa-Fc binding on should be assessed on EV-exposed KO cells.

The assay could be performed using additional cross-linking of SIRPa-Fc protein by anti-Fc Fab'2 or plate-bound SIRPa-Fc protein.

Why cross-dressing does not happen between Jurkat WT and CD47ko, thereby sensitizing the CD47ko Jurkat to SIRPa-induce apoptosis?

Complementation of CD47ko clones with CD47 would provide a nice control ruling out possible artefacts of CRISPR targeting.

6) PMID: 28053997 needs to be properly cited, and the author's need further data to support their claim that "hCD47 cross-dressing is SIRPα-independent". Specifically, they need to quantify 'cross-dressing' by non-phagocytic cells that express a suitable hSIRPa in the presence or absence of SIRPa blocking Ab.

7) While Figure S5 nicely shows that SIRPa-Fc causes apoptosis on the cells that express CD47 in a dose-dependent manner, the actual Figure 5 fails to support the claim that EV-based crossdressing protects the cells from apoptosis. Crossdressing doesn't seem to cause a notable increase in hCD47 expression on R1 cells, so the difference in apoptosis between R1 and R2 is likely due to the comparably low expression of CD47 in R1 despite the EV treatment.

---

## [Author Response]

Essential revisions:1) Figure 1The rationale for the use of CD47 isoforms 2 and 4 requires further detail a full explanation.

There are 4 alternatively spliced CD47 isoforms that differ from each other at their intracytoplasmic carboxy termini (Reinhold et al., 1995). Among these isoforms, isoform 2 is most widely expressed in all cells, including both hematopoietic and nonhematopoietic cells, and isoform 4 is the isoform with longest intracellular domain. Because the difference among 4 CD47 isoforms is at the intracytoplasmic carboxy termini, these isoforms are all capable of interacting with SIRPα to inhibit phagocytosis. The purpose to test two isoforms was to rule out the possibility that our findings are isoform-specific, and choosing isoforms 2 and 4 were because of their relatively unique features (most wide expression for isoform 2 and longest intracytoplasmic tail for isoform 4), and both were used for making transgenic expression of human CD47 to prevent rejection (Ide et al., 2007, Hosny et al., 2021). We have now pointed this in the revised manuscript (in the Results section).

Please provide kinetics or the dynamic of CD47 acquisition in recipient cells.

The data presented in the previous manuscript (Figure 3A and 3B) showed that the level of CD47 cross-dressing on recipient cells remained unchanged from 2 to 6 hours. New experiments were performed to further determine the kinetics of CD47 acquisition, in which pig LCL cells were cultured with EVs from hCD47-transgenic LCL cells or WT Jurkat cells. We found that the process of CD47 cross-dressing is fast, significant CD47 acquisition was detected 1 hour, and the levels remained stable or increased at 24 hours after culturing with EVs (Figure 3—figure supplement 1 in the revised manuscript).

Is cross-dressing blocked at 4deg?

To address this question, we performed a new experiment to compare CD47 crossdressing on pig LCL cells that were incubated with EVs of hCD47-LCL cells at 37°C or 4°C, and the results showed clearly that CD47 cross-dressing was significantly blocked at 4°C (Author response image 1).

**Author response image 1. sa2fig1:** CD47 cross-dressing is significantly blocked at 4 °C. EVs from hCD47-LCL cells were labeled with PKH26, and recipient WT pig LCL cells were labeled with PKH67, EVs and recipient cells were cocultured for 6h and washed, then analyzed for PKH26 (A) and hCD47 (B) cross-dressing on recipient cells by flow cytometry using anti-hCD47-AF647 mAb.Results shown are representative of three independent experiments.

Does cross-dressing require cell-to-cell contact?

Cell-to-cell contact is not required for cross-dressing, as CD47 cross-dressing also occurred when cells were incubated with EVs or Exos.

Does cross-dressing require a functional cytoskeleton?

To address this question, we performed new experiments to determine whether CD47 cross-dressing can be blocked by treatment with Cytochalasin D, a cell-permeable potent inhibitor of actin polymerization that disrupts actin microfilaments. We found that CD47 cross-dressing was clearly detected on cells pre-treated with Cytochalasin D, but to lower levels compared to untreated cells. The results indicate that a functional cytoskeleton is not absolutely required for, but may facilitate CD47 cross-dressing. We have now presented these new data in the revised manuscript (Figure 2—figure supplement 3 in the revised manuscript).

Do other membrane proteins get transferred as well or is it a peculiarity of CD47h2/h4 to be embedded in cross-dressing vesicles released from donor cells?

Other membrane proteins on EVs should be detected on recipient cells cross-dressed by EVs, and it is unlikely that EVs express only CD47. In support of this possibility, our new data showed that, not only hCD47, but also hCD45, hCD90 and HLA-ABC proteins were detected on pig LCL cells after incubated with WT Jurkat cells. We have now pointed this out in the revised manuscript (Figure 2—figure supplement 2).

Does cross-dressed CD47 resist to EDTA washes (one might anticipate that tethered CD47 would not resist while CD47 actually transferred by membrane fusion would)?

New experiments were performed to address this question, in which donor and recipient cells that were labeled with different fluorescence dyes and cultured for 24 hours, following by washing with PBS buffer or PBS buffer containing trypsin/EDTA, then analyzed gated recipient cells for fluorescence acquired from the donor cells. We found that treatment with trypsin/EDTA did not reduce the fluorescence intensity, indicating that membrane fusion is involved in CD47 cross-dressing. We have now included these data and discussed this issue in the revised manuscript (Figure 3—figure supplement 2).

What is the persistence of crossdressed CD47 on recipient cells (post-separation from donor cells)?

To address this question, CD47-transgenic LCL or WT Jurkat cells were labeled with PHK26 and co-cultured with PKH67 labeled pig LCL cells for 1 day, then recipient cells (PKH67+) are sorted and treated with mitomycin C (2ug/ml) for 30min (to prevent dilution of cross-dressed proteins by cell division), and cultured in media to determine the persistence of cross-dressed hCD47. Although the level (MFI) of hCD47 on recipient LCL cells gradually decreased, >50% remained 3 days after culturing, indicating that CD47 cross-dressing is quite stable and may persist for several days. We have now pointed this out in the revised manuscript (Figure 2C).

b) i) Why "ligation of the autogenous, but not cross-dressed, CD47 induced cell death."

Autogenous CD47 contains transmembrane and intracytoplasmic domains that are required for transferring death signals. Although this study does not tell the structure of hCD47 cross-dressed on the recipient cells, the failure of cross-dressed CD47 to induce apoptosis indicates that cross-dressed CD47 cannot effectively deliver the intracytoplasmic signaling, leading to activation of apoptotic signal cascades.

(ii) Please show the extracellular vesicles and/or exosomes, do or do not transfer hCD47 mRNA or even DNA (e.g. plasmid, even episomal).

We have analyzed and found that the EVs contain hCD47 DNA and RNA, as well as housekeeping gene GAPDH RNA (Author response image 2). Although we do not have direct evidence, these data suggest that DNA and RNA within EVs might also be transferred to the recipient cells. Indeed, unlike protein delivery that can be tracked by fluorescent dye labeling, progress on understanding DNA and RNA cargo delivery by EVs has been hampered by the lack of efficient and specific tracking methods(O’Brien et al., 2020). We agree that elucidating this issue, including the efficacy of DNA and RNA transfer, their persistence and functional significance, will be helpful in understanding the insights into EV cross-dressing, and would like to perform these studies in the future. Nonetheless, colocalization of CD47 staining and cell-tracker dye fluorescence on the recipient cells (Author response image 3) confirmed that the majority of CD47 on the recipient cells were cross-dressed CD47 proteins and not derived from DNA or RNA cargo conveyed in EVs.

**Author response image 2. sa2fig2:** EVs from pig LCL/CD47^p/h^ cells contain both hCD47 DNA and RNA, and GAPDH RNA. DNA and total RNA are extracted from hCD47 transgenic pig LCL and their MVs, and WT pig LCL cells respectively, then cDNA was synthesized from total RNA using a reverse transcription kit, and hCD47 and GAPDH sequences are amplified by PCR. Shown are Agarose gel electrophoresis of hCD47 genomic DNA (A), hCD47 cDNA (B) and GAPDH cDNA PCR products..

**Author response image 3. sa2fig3:** Colocalization of CD47 proteins and EVs on pig LCL cells. PKH26-labeled EVs from LCL/CD47^p/h^ cells were co-cultured with pig LCL cells for 6h, then stained by anti-huCD47-AF647 antibody and analyzed by confocal microscope. Shown are CD47 staining (blue; left), EVs (PKH26 red fluorescence, mid) and overlayed (right) images.

(iii) Determine the reason(s) and implications for function (e.g. phagocytosis) for the very broad distribution of intensities on cells that supposedly have 'all' received hCD47.

Although the distribution of cross-dressed CD47 is relatively broader than autogenous CD47, the distribution patterns in most experiments were similar between these cells. However, different patterns were seen in some experiments (Figure 1E; Figure 3). While we do not fully understand the reasons, the difference may possibly be due to variable hCD47 on the donor cells for the experiment in Figure 1E, in which primary hCD47-tg pig BM cells that express highly variable levels of hCD47 were used as the donor cells; and for the experiments shown in Figure 3, insufficient EVs or Exos in cultures might be attributable to it. Furthermore, functional heterogeneity in proliferation and senescence may also affect the efficacy of crossdressing. Nonetheless, we agree that the level of CD47 cross-dressing should have an impact on the degree of protection against phagocytosis.

(iv) Show whether "florescence Celltrace violet" transfers from labelled cells to unlabeled cells (Figure 1c).

It is expected that fluorescence dye that labeling membrane proteins should also be transferred with EVs from labeled cells to unlabeled cells, but it was difficult to see it in Figure 1C, in which no negative controls (i.e., uncultured cells) were analyzed at the same time. To clarify this, we performed a new experiment, in which we used the cells that were mixed before flow cytometry analysis as the negative control. As shown in Author response image 4, fluorescence dye transfer was clearly detected in co-cultured cells compared to the controls (i.e., a mixture of labeled and unlabeled cells that were mixed before flow cytometric analysis), but the labeled cells could be easily separated from cross-dressed unlabeled cells on flow cytometric plots. In these experiments, we also stained the cells with anti-hCD47 antibody, and the results confirmed that both fluorescence dye and anti-hCD47 staining can clearly separate the donor cells from cross-dressed recipient cells, and the results obtained by these methods are identical (this also address the reviewer’s comment #1c below). Furthermore, anti-CD47 staining and fluorescence dye used are in different channels (i.e., different colors), so fluorescence dye-crossdressing is not a confounding factor for the experiments in which cells were also stained by anti-hCD47 antibody.

**Author response image 4. sa2fig4:** Florescence dye cross-dressing. PKH26-labeled hCD47-LCL (A) or PKH26labeled WT Jurkat (B) as donor cells were co-cultured, respectively, with PKH67-labeled pig LCL for 24h, and then analyzed for fluorescent dye cross-dressing by flow cytometry using anti-hCD47AF647 mAb. Two types of cells mixed immediately prior to flow cytometry analysis (without co-culture) were used as negative controls. Shown are flow cytometry profiles (left panels) and median fluorescence intensities (right panels). Results shown are representative of two independent experiments.

(v) Determine why results for h2 and h4 spliceforms are variable; sometimes the same (e.g. Figure 1a) versus sometimes different (Figure 1b).

As described in the manuscript, different donor cells were used in the experiments shown in Figure 1A and Figure 1B. In Figure 1A, the donor cells were PAOCs that were genetically modified to express hCD47 isoform 2 (i.e., expressed both pig and human isoform 2 CD47 and referred to as PAOC/CD47^p/h2^) or isoform 4 (i.e., i.e., expressed both pig and human isoform 4 CD47 and referred to as PAOC/CD47^p/h4^). However, in Figure 1B, the donor cells were CD47defficient PAOC (PAOC/CD47^null^) cells that were genetically modified to express transgenic hCD47 isoform 2 (PAOC/CD47^h2^) or isoform 4 (PAOC/CD47^h4^), which express only human CD47 (isoform 2 or isoform 4) but no pig CD47. It is perceivable that different transgenic cell lines may express different levels of the transgene. Indeed, the results for each cell line were considerably stable throughout the experiments.

c) In Figure 1A-B: How did the authors gate on the PAOCp in the coculture? Congenic marker, cell tracker? 1C is the right way to do it as surface markers/ membrane stains can't be relied upon. The histograms may be misleading as the shifted curves can include both the acceptor cells that capture CD47h and donor cells that had reduced CD47h. 1D: It's not possible to say that it's fully SIRP-a independent as the increase in the CD47 cross-dressing looks more modest compared to 1-A,B,C. However, this may be due to an intrinsic difference between the starting levels of CD47 on tg LCL and PAOC. Yet, this reviewer recommends changing the statement at line 131 as such: "…., indicating that hCD47 cross-dressing can occur in a SIRPα-independent manner."

We agree with the reviewer, and that is why we performed the experiment in Figure 1C to validate the results shown in Figure 1A-B. As we discussed in the manuscript (line 106-112 in the revised manuscript), “To rule out the possibility that, during the co-culture, the CD47^null^ cells did not become CD47+, but PAOC/CD47^h2^ cells reduced hCD47 expression, we labeled PAOC/CD47^null^ (Figure 1C, top) or PAOC/CD47^h2^ (Figure 1C, bottom) cells with florescence Celltrace violet, and then cocultured the labeled cells with unlabeled PAOC/CD47^h2^ or PAOC/CD47^null^ cells, respectively. This experiment, in which fluorescence-labeling allowed for better distinguishing between the two cell populations in the cocultures, further confirmed that PAOC/CD47^null^ cells can be cross-dressed by CD47 after coculture with PAOC/CD47^h2^ cells (Figure 1C).” The results from the experiment in Figure 1C confirmed that PAOC^null^ and PAOC^h2/h4^ cells can be clearly separated by anti-hCD47 antibody staining. This conclusion is also supported by the results of our new experiments that were performed to address the reviewer’s comments on fluorescence dye transfer (Comment #1b-iv).

We agree with the reviewer that our data cannot rule out the possibility of SIRPα-mediated crossdressing, and therefore, as the reviewer suggested, revised our statement to “Although we cannot rule out of the role of SIRPα in hCD47 cross-dressing, our data indicate that hCD47 cross-dressing can occur in a SIRPα-independent manner” in the revised manuscript.

2) Figure 2In Figure 2 Please address labelling and gating the cells in the coculture (see comments 1c).

In the experiments shown in Figure 2, CD47KO Jurkat and pig LCL recipient cells were gated by anti-hCD47 staining. As discussed in our responses to Comment #1c and #1b-iv above, we have firmly confirmed that anti-hCD47 staining can clearly distinguish cross-dressed recipient cells from donor cells by using fluorescence dye labeling as the reviewer suggested.

3) Figure 3The titration of EVs/Exos should be performed to highlight the efficiency of the cross-dressing process.The yield of EV/Exos production is explained in order to get an idea if the concentration/quantity of EVs correspond to a reasonable amount of cells and how this number compares to the number of cross-dressed "recipient" cells.Does membrane acquisition upon EV/Exos involve membrane fusion?(this can be addressed using fluorescent membrane probes).

We thank the reviewer for pointing out this question. In our study, we used a simple centrifugation method to prepare EVs and Exos, which may not efficiently recover entire and pure EVs or Exos, so the results may not be very helpful in addressing the reviewer’s question. However, in the cell co-culture experiments, the donor to recipient cell ratios ranged from 1:1 to 1:3, which are considered reasonable and can be expected to occur in real in vivo situation, as for both organ/cell transplantation and cancer conditions, the numbers of graft- or tumor-infiltrating recipient cells are expected to be less than the numbers of graft or tumor cells. Thus, our results are of clinical significance.

The results of our new experiments indicate that membrane fusion is involved in membrane acquisition of EVs/Exos. We found that treatment with Trypsin/EDTA did not reduce crossdressed fluorescence dye in the recipient cells. We have now included these new data and discussed this issue in the revised manuscript (Figure 3—figure supplement 2).

In Figure 3C: Why did the authors cocultured cells for 18 and 42 hours? At 42h PAOC cells line would be expected to divide and dilute the CTV, why didn't they? Also, the fitness of the cells might be impaired at the late time points, have the authors performed a viability staining and gated out the dead cells?

PAOC^null^ cells are adherent endothelial cell line cells, and their proliferation is slower than tumor cell lines used (i.e., Jurkat and LCL cells). We found that EV acquisition by POAC cells is slower than Jurkat and LCL cells, assumably due to their different properties (for example the adherent property) and therefore, these cells were cultured for a longer period than Jurkat and LCL cells. The relatively lower hCD47 staining at 42 hours than 18 hours may be, at least partially, attributed to dilution by cell proliferation, as pointed out by the reviewer. Nonetheless, hCD47 cross-dressing was clearly detected in POAC cells in our experiments, and the data support our conclusion. We have confirmed that PAOC cells were in good conditions with 95% alive at 48h.

Also, dead cells were gated out by PI or 7-AAD staining in all flow cytometry analysis.

4) Figure 4It is unclear if CD14+CTV+ events all correspond to phagocytosis or doublets formation upon tethering of both cell types. Imaging experiments using confocal microscopy on sorted CD14+CTV+ events could easily quantify this. Please add quantitative imaging data.This assay requires rigorous viability staining, a robust gating strategy to identify the donor and acceptor cells of the co-culture and controls with single cells. Without those, how do the authors claim that the CTV+ cells in the gate are monocytes that had phagocytosed the CTV-labeled cells?

This flow cytometry-based phagocytosis assay is well established and used routinely in the lab, in which all dead (PI+ or 7-AAD+) cells are excluded from the first gating, then doublets are excluded by gating on FSC-A vs FSC-H plots (Author response image 5). This method has also been validated by imaging experiments using confocal microscopy as the reviewer suggested, confirming that CVT+CD14+ cells are macrophages that have engulfed CTV+ target cells (Author response image 6).

**Author response image 5. sa2fig5:** Gating strategy for analyzing phagocytosis by macrophages. All PI+ cells are excluded from the first gating, then doublets are excluded by gating on FSC-A vs FSC-H. Macrophages were gated on FSC-A vs SSC-A, then by CD45 and CD14 expression.

**Author response image 6. sa2fig6:** Confocal microscopic analysis of phagocytosis. CFSE-labeled CD47KO Jurkat cells (green) were cocultured with human macrophages for 2h, floating cells were washed out by PBS, then macrophages were harvested by trypsin/EDTA digestion and stained by anti-huSIRPα-APC (red) antibody and DAPI (blue). The samples were subjected to confocal laser scanning microscopy, and the results confirmed that target cells were engulfed by human macrophages. Shown are representative confocal images.

4B: The population sorted is likely a mix of donor and acceptor cells and contains CD47 expressing cells too. This should have been carefully carried out by labeling the cells with cell trackers prior to culture and sorting them according to the label.

As discussed in our responses to the reviewers’ comments (#1b-iv and #1c) above, we have validated this method using fluorescent CTV labeling and confirmed that anti-hCD47 staining can reliably separate donor cells from recipient cells with hCD47 cross-dressing. Thus, the gated population should have extremely low contamination by donor cells. Indeed, this was also confirmed by the donor (PAOC47^h2^) cell alone control included in this experiment. As shown in Figure 4B (middle), there were only <1% of cells in the gated region (R2) in the cultures of PAOC47^h2^ only.

4C: What does the CD14 staining of PAOC cells before and after coculture look like? In systems where cross-dressing occurs, it's hard to rely on the surface markers as they have mobility between cells. Have the authors considered labelling donor and acceptor cells with two different cell trackers to set the gates properly and independent of surface markers?

We have confirmed that anti-human CD14 antibody does not cross-react with pig cells, and POAC cells after co-cultured with human macrophages remained negative relevant to human macrophages (Author response image 7).

**Author response image 7. sa2fig7:** Anti-human CD14 antibody does not cross-react with PAOC cells. PAOC cells were cocultured with human macrophages and analyzed for human CD14 and CD45 expression. Representative flow cytometry profiles showing staining of anti-hCD14 and anti-hCD45. The results conformed that after coculture, POAC cells and human macrophages could be clearly separated by staining with antihuman CD14 and CD45 antibodies on flow cytometry.

5) Figure 5SIRPa-Fc binding on should be assessed on EV-exposed KO cells.

We have assessed SIRPα-Fc binding on EV-exposed CD47KO Jurkat cells as the reviewer suggested (Figure 4A). We also assessed phagocytosis by human macrophages (Figure 4) and SIRPα-Fc-induced apoptosis (Figure 5E-F) of EV-exposed KO cells.

The assay could be performed using additional cross-linking of SIRPa-Fc protein by anti-Fc Fab'2 or plate-bound SIRPa-Fc protein.

Although we do not have direct evidence to support this, we would agree with the reviewer that additional cross-linking may possibly enhance apoptosis. Since SIRPα-Fc without additional cross-linking induced significant apoptosis in our experiments, we did not perform any experiments using additional cross-linking.

Why cross-dressing does not happen between Jurkat WT and CD47ko, thereby sensitizing the CD47ko Jurkat to SIRPa-induce apoptosis?

The purpose of this experiment (Figure 5C) was to determine whether SIRPα-Fc may induce apoptosis of CD47KO cells without hCD47 cross-dressing (as controls for the experiment shown in Figure 5E, 5F). In this experiment, two types of cells were cocultured for only 1h in the presence of SIRPa-Fc proteins, so there should be very minimal cross-dressing during the culture period. For determining whether hCD47+ EV cross-dressing may induce apoptosis, CD47KO cells were pre-exposed to hCD47+ EVs before mixed with WT cells for induction of apoptosis by SIRPa-Fc (Figure 5E, 5F) and the results showed that CD47 cross-dressing does not transmit death signaling induced by SIRPα-Fc.

Complementation of CD47ko clones with CD47 would provide a nice control ruling out possible artefacts of CRISPR targeting.

We thank the reviewer for the thoughtful suggestion. To address this comment, we transfected CD47KO Jurkat cells with CD47, and found that CD47KO Jurkat cells transfected with CD47 became susceptible to apoptosis induced by SIRPα-Fc. We have now included these data in the revised manuscript (Figure 5—figure supplement 3).

6) PMID: 28053997 needs to be properly cited, and the author's need further data to support their claim that "hCD47 cross-dressing is SIRPα-independent". Specifically, they need to quantify 'cross-dressing' by non-phagocytic cells that express a suitable hSIRPa in the presence or absence of SIRPa blocking Ab.

We thank the reviewer for bring this reference to our attention (which has now been cited in the revised manuscript), and we agree that the interaction of CD47 with SIRPα may facilitate transduction of SIRPα+ non-phagocytic cells with CD47-Lenti. In our study, hCD47 cross-dressing occurred in cells that do not express SIRPα, indicating cross-dressing can occur independent of SIRPα. However, our data cannot rule out the role of SIRPα in hCD47 crossdressing, and therefore, we have revised our statement of "hCD47 cross-dressing is SIRPαindependent" to “hCD47 cross-dressing can occur in a SIRPα-independent manner." (Please refer to our response to a similar comment raised by another reviewer (see Comment #1c above)).

7) While Figure S5 nicely shows that SIRPa-Fc causes apoptosis on the cells that express CD47 in a dose-dependent manner, the actual Figure 5 fails to support the claim that EV-based crossdressing protects the cells from apoptosis. Crossdressing doesn't seem to cause a notable increase in hCD47 expression on R1 cells, so the difference in apoptosis between R1 and R2 is likely due to the comparably low expression of CD47 in R1 despite the EV treatment.

As we explained in the manuscript that the low staining of hCD47 in Figure 5 was due to the fact that prior incubation with hSIRPα-Fc proteins can block subsequent staining by antihCD47 mAb (data are presented in the previous Figure S5, which is Figure 5—figure supplement 2 in the revised manuscript). Please see page 15 of the manuscript, where we point out that “Of note, binding of hSIRPα-Fc proteins to cell surface CD47 (either native or cross-dressed) could block subsequent staining with anti-hCD47 antibodies and thus, the cells cultured with hSIRPα-Fc showed relatively lower hCD47 staining than those cultured without (previous Figure S5, which is Figure 5—figure supplement 2 in the revised manuscript, Figure 5).” Data in Figure 4A also showed that EV treatment resulted in a significant hCD47 cross-dressing.

References

Hosny, N., Matson, A. W., Kumbha, R., Steinhoff, M., Sushil Rao, J., Elabaseri, T. B., Sabek, N. A., Mahmoud, M. A., Hering, B. J. & Burlak, C. 2021. 3'UTR enhances hCD47 cell surface expression, self-signal function, and reduces ER stress in porcine fibroblasts. *Xenotransplantation,* 28**,** e12641.

Ide, K., Wang, H., Liu, J., Wang, X., Asahara, T., Sykes, M., Yang, Y. G. & Ohdan, H. 2007. Role for CD47-SIRPà signaling in xenograft rejection by macrophages. *Proc Natl Acad Sci USA,* 104**,** 5062-5066.

O’brien, K., Breyne, K., Ughetto, S., Laurent, L. C. & Breakefield, X. O. 2020. RNA delivery by extracellular vesicles in mammalian cells and its applications. *Nature Reviews Molecular Cell Biology,* 21**,** 585-606.

Reinhold, M. I., Lindberg, F. P., Plas, D., Reynolds, S., Peters, M. G. & Brown, E. J. 1995. in vivo expression of alternatively spliced forms of integrin-associated protein (CD47). *J Cell Sci,* 108 ( Pt 11)**,** 3419-25.